# Quantitative Analyses and Validation of Phospholipids and Sphingolipids in Ischemic Rat Brains

**DOI:** 10.3390/metabo12111075

**Published:** 2022-11-06

**Authors:** Chiung-Yin Huang, Ping-Ju Tsai, Hsuan-Wen Wu, I-Ting Chen, Hay-Yan J. Wang

**Affiliations:** 1Neuroscience Research Center, Chang Gung Memorial Hospital, Taoyuan 333012, Taiwan; 2Department of Neurosurgery, New Taipei Municipal TuCheng Hospital, New Taipei City 236027, Taiwan; 3Department of Biological Sciences, National Sun Yat-Sen University, Kaohsiung 804201, Taiwan; 4Department of Surgery, Yuan’s General Hospital, Kaohsiung 802635, Taiwan

**Keywords:** ischemic stroke, hydrophilic interaction chromatography–tandem mass spectrometry, phospholipids, sphingolipids, multivariate data analyses, tissue lipid biomarkers, lipidomics

## Abstract

Prior MALDI mass spectrometry imaging (MALDI-MSI) studies reported significant changes in phosphatidylcholines (PCs), lysophosphatidylcholines (LPCs), and sphingomyelins (SMs) in ischemic rat brains yet overlooked the information on other classes of PLs and SLs and provided very little or no validation on the detected lipid markers. Relative quantitation of four classes of PLs and two classes of SLs in the ischemic and normal temporal cortex (TCX), parietal cortex (PCX), and striatum (ST) of rats was performed with hydrophilic interaction chromatography (HILIC)–tandem mass spectrometry (MS/MS) analyses, and the marker lipid species was identified by multivariate data analysis and validated with additional tissue cohorts. The acquired lipid information was sufficient in differentiating individual anatomical regions under different pathological states, identifying region-specific ischemic brain lipid markers and revealing additional PL and SL markers not reported previously. Validation of orthogonal partial least square discriminating analysis (OPLS-DA) identified ischemic brain lipid markers yielded much higher classification accuracy, precision, specificity, sensitivity, and lower false positive and false negative rates than those from the volcano plot analyses using conventional statistical significance and a fold change of two as the cutoff and provided a wider prospective to ischemia-associated brain lipid changes.

## 1. Introduction

Cerebrovascular accident (CVA), also known as stroke, is caused by the interruption or significant reduction of blood supply to the brain parenchyma. Based on the nature of the vascular event leading to the compromise of blood supply, stroke is generally categorized as ischemic stroke, which accounts for approximately 80% of vascular incidents, or hemorrhagic stroke [1,2]. Approximately 20–30% of the ischemic stroke patients were eventually associated with atrial fibrillation [3]. The extent of loss of brain function by stroke largely depends on the location of the precipitated vascular event, the volume of the brain parenchyma affected by such event, and the duration of blood flow interruption. Both ischemic and hemorrhagic stroke provoke critical adverse cellular responses in neurons that will eventually hamper the mobility and physiological capabilities of the affected subjects to various extents, which, when severe, would in turn negatively impact the psychological and socioeconomical well-being of the patients and their immediate families.

Various pathological events are triggered, amplified, and progressed in the brain parenchyma following the initial stoppage of blood flow in stroke. The cell damage and death by the immediate compromise of blood supply triggers the release of inflammatory mediators such as interleukins (ILs), tumor necrosis factors (TNFs), cytokines, phospholipase, and other inflammatory mediators, among others, that would further perpetuate the inflammatory responses in the brain tissue [2,4,5,6,7]. Earlier studies demonstrated that the inhibition of cyclooxygenase 2 (COX 2) activity in the ischemic brain attenuated the parenchymal damage, reduced the extent of brain edema, leukocyte infiltration, and apoptosis [8,9,10,11,12]. Such observation strongly indicated the critical roles of lipid metabolism in the manifestation of ischemia-mediated brain injuries. Conceivably, revealing the changes of lipid composition in the ischemic brain tissue may unveil previously overlooked opportunities to reduce ischemia-mediated brain damage. Several matrix-assisted laser desorption–ionization mass spectrometry (MALDI-MS) profiling and imaging studies have surveyed the changes of phospholipids (PLs) and sphingolipids (SLs) in ischemic rodent brains [13,14,15,16]. These studies reported changes of commonly detected phosphatidylcholines (PCs), lysophosphatidylcholines (LPCs), and sphingomyelins (SMs) in the ischemic brain parenchyma with histological distribution, especially from studies of MALDI-MS imaging (MALDI-MSI). However, very few studies reported changes of other highly abundant PLs and SLs in the ischemic brain. The incomplete coverage of lipid information in the ischemic brain parenchyma would bias the identification of lipid markers in brain ischemia toward candidates preferentially detected by the MALDI method and omit the contribution of other PLs and SLs not easily observed under the same detection method.

To address the discrepancy of lipidomic information in the ischemic brain parenchyma, we analyzed the commonly encountered PLs and SLs from the infarcted rat brains by LC-MS/MS with the use of internal standards. The tissue sampling regions included the ischemic temporal cortex (TCX) that corresponded to the core of ischemia by middle cerebral artery occlusion (MCAO) approach in rats, the parietal cortex (PCX) close to the margin of ischemia that was deemed as the penumbra of ischemia, and the striatum (ST) corresponding to the subcortical ischemic brain region. Lipids in the respective contralateral nonischemic brain regions were also analyzed for comparison. Multivariate data analyses were performed on the acquired lipid information to identify the potential lipid markers of brain tissue at different anatomical regions, with subsequent validation through modeling and classification of additional animal cohorts.

## 2. Materials and Methods

### 2.1. Chemicals

HPLC-grade ammonium formate and LC-MS-grade formic acid were purchased from Fluka Chemie (Buchs, Switzerland). LC-MS-grade acetonitrile (ACN), methanol (MeOH), and ACS-grade chloroform were purchased from J.T. Baker or Mallinckrodt (Mallinckrodt Baker Inc. Phillipsburg, NJ, USA). 2,6-di-tert-butyl-4-methylphenol (butylated hydroxytoluene; BHT) was purchased from Sigma Chemical Co. (St. Louis, MO, USA). 2,3,5-triphenyl-tetrazolium chloride (TTC) was purchased from Alfa Aesar (Lancashire, U.K.). Water (18.2 MΩ/cm) was purified in house using a Synergy Ultrapure Water System (Millipore Co., Burlington, MA, USA). The following lipid standards were purchased from Avanti Polar Lipids Inc. (Alabaster, AL, USA): 1-myristoyl-2-hydroxy-sn-glycero-3-phosphocholine (LPC 14:0), 1-myristoyl-2-hydroxy-sn-glycero-3-phosphoethanolamine (LPE 14:0), 1,2-dimyristoyl-sn-glycero-3-phosphocholine (PC 14:0/14:0), 1,2-dimyristoyl-sn-glycero-3-phosphoethanolamine (PE 14:0/14:0), N-lauroyl-D-erythro-sphingosylphosphorylcholine (SM d18:1/12:0), and N-heptadecanoyl-D-erythro-sphingosine (ceramide d18:1/17:0).

### 2.2. Animal Handling and Tissue Collection

All the animal care and use was in accordance with the U.S. Public Health Service Policy on Humane Care and Use of Laboratory Animals. The animal experiment protocols (No. 99-001 and 105-34) were approved by the Institutional Animal Care and Use Committee (IACUC) of National Sun Yat-Sen University. Male Sprague Dawley rats between 200 and 250 g were purchased from BioLASCO Taiwan Co. Ltd., Taiwan, and group-housed in the colony room under a 12-h dark–light cycle. Animals with body weight between 280 g and 350 g were used for study. Surgical induction of permanent middle cerebral artery occlusion (pMCAO) to create ischemic stroke and the subsequent neurological evaluation and verification followed the previously described protocol [13]. Modification of nylon filament used in the MCAO induction followed the method of Spratt et al. [17].

Twenty-four hours after the successful pMCAO, rats were euthanized with isoflurane and decapitated immediately after breathing stopped. The brains were rapidly dissected from the cranium and placed in 4 °C normal saline solution for 2–3 min to wash out the excessive blood, then placed in a brain matrix (SA-2160; Roboz Surgical Instrument Co., Gaithersburg, MD, USA). Then, the entire brain was cut between the anterior and posterior pole of cerebrum into 2 mm thick coronal slices. The fourth section from the anterior cerebral pole corresponding to 0.2 mm rostral of bregma [18] was chosen for tissue puncture. The punctured brain tissue was individually weighed and placed in a new glass sample vial. Thereafter, the brain slice was stained with 0.1% TTC solution under 37 °C for 15 min and fixed in 10% buffered formalin. The image of the stained brain slice was scanned with a flatbed scanner for further analysis.

### 2.3. Tissue Lipid Extraction

Lipids in the punctured brain tissue were extracted using the method of Bligh and Dyer [19] with the modification of Ivanova et al. [20] The punctured brain tissue was placed in a 2 mL glass sample vial containing a mixture of 250 µL methanol (MeOH), 250 µL of 0.1 N HCl, and a suitable amount of lipid internal standards (LIS; see below). Then, the tissue was thoroughly dispersed and homogenized for 1 min. A 10 µL aliquot of LIS containing 2.16 nmole of PE 14:0/14:0, 1.20 nmole of LPE 14:0, 2.16 nmole of PC 14:0/14:0, 1.08 nmole of SM d18:1/12:0, 1.73 nmole of LPC 14:0, and 1.81 nmole of ceramide d18:1/17:0 was added to the methanol/HCl mixture for every 3 mg of brain tissue. After the initial homogenization, 1 mL of chloroform was added to the homogenate and the sample vial cap-sealed with a PTFE septum and vortexed for 15 min under room temperature. The mixture was briefly centrifuged (3500× *g*, 10 min, 4 °C). Approximately 750–800 µL of lower organic layer was aspirated and collected in a new glass sample vial and dried under a gentle stream of nitrogen until complete dryness. The dried lipid residue was sealed in nitrogen and stored under −80 °C until LC-MS/MS analysis. Immediately prior to LC-MS/MS analysis, 750 µL of mobile phase A (see below) was added to the sample vial containing lipid residue, briefly vortexed to dissolve the lipids, and then queued in the autosampler at 4–6 °C for analyses.

### 2.4. Hydrophilic Interaction Chromatography–Tandem Mass Spectrometry

The hydrophilic interaction chromatography (HILIC)–tandem mass spectrometry (MS/MS) system consists of a Waters 2695 HPLC separation module coupled to an AmaZon X ion trap mass spectrometer (Bruker Daltonics, Bremen, Germany). The chromatographic separation of lipids was carried out on a Supelco Ascentis^®^ Express HILIC column (2.1 mm × 150 mm; 2.7 µm; 53946U, Sigma-Aldrich Co. USA) coupled with a guard cartridge (53520U, Sigma-Aldrich Co.). The column temperature was maintained at 30 °C for lipid separation. The mobile phases were modified from a previously reported method [21] where mobile phase A consisted of 85% ACN, 10% MeOH, and 5% H_2_O (*v*/*v*/*v*), and mobile phase B was composed of 65% ACN, 10% MeOH, and 25% H_2_O. Both mobile phases contained 1 mM ammonium formate and 0.04% (*v*/*v*) formic acid. The mobile phase was delivered at 0.2 mL/min, using the following gradient for lipid class elution: mobile phase A was initially held at 90% for 6 min, then linearly decreased to 50% in 1 min, then held at 50% for the next 11 min, then linearly increased to 90% in 1 min, and then held at 90% till the end of the run at 35 min.

Outflow of the HILIC column was connected to the electrospray ionization (ESI) nozzle of the mass spectrometer to ionize the eluted lipids. The mass spectrometer was operated under multiple-reaction monitoring (MRM) mode using Ultrascan mode, setting the 12C ± 0.5 Da as the mass isolation window for tandem mass spectrometry. The maximum trap accumulation time was 200 ms and the Ion Charge Control was set to 100,000. Five µL of the reconstituted lipid sample was injected onto the HILIC column for LC-MS/MS analysis. An injection of mobile phase A, separated with a full gradient run, was placed between two lipid samples to avoid carryover. Each lipid sample was analyzed twice with two different sets of MS/MS methods under the same chromatographic gradient to cover the lipid species detected by this platform. Detailed MS/MS parameters are listed in Appendix A. Relative lipid quantitation was performed by calculating the ratio of the area under the chromatographic peak from the quantitative fragment of individual lipid to that of its respective LIS. To ensure a reliable relative quantitation, the ratio of the sample lipid to LIS was bracketed between 0.1 and 10 [22]. The quantitative fragmentation scheme followed a previously reported method [23] after slight modification where [M+HCOO^−^−60]- fragment ions were monitored for the quantitation of PCs, LPCs, and SMs, while the identities and intensities of the fatty acid fragment ions were used to identify the fatty acyl composition of the precursor. The [M+H-141]+ fragment ions were monitored for the quantitation of PE and LPEs, and the fatty acid information was acquired with an additional run under negative ion mode to identify the PE species. The *m*/*z* 264.3 fragment ion from MS3 of the protonated ceramide precursors was monitored for quantitation [24].

### 2.5. Data Analysis

Multivariate data analysis (MVDA) was carried out by SIMCA 14.1 (Umetrics, Umeå, Sweden) and MetaboAnalyst 5.0 (https://www.metaboanalyst.ca/; access date: 20 June 2020–31 July 2022) [25]. Calculation of the descriptive statistics such as the t-statistic values, the *p* values of unpaired *t*-test, and the multiple-sample receiver operating characteristic curve (MSROCC) modeling analysis and the area under curve (AUC) calculation were carried out using MetaboAnalyst 5.0 and GraphPad Prism 8.4.3 (GraphPad Software, San Diego, CA, USA). MSROCC model-based biomarker analyses and new sample classification were carried out using MetaboAnalysis 5.0. The accuracy, precision, sensitivity, specificity, false positive rate, and false negative rate of the lipid marker panels identified by different MVDA methods were calculated by the confusion matrix [26].

## 3. Results

### 3.1. Histology

The brain slice sampled for lipid analysis is shown in Figure 1. The normal left (red) and ischemic right (white) brain areas were revealed by the TTC stain together with the white matter region, such as the corpus callosum, showing its lack of staining even in the normal hemisphere. The sampled regions in the ischemic (I-) and normal (N-) temporal cortex (TCX), parietal cortex (PTX), and striatum (ST) were labeled accordingly. The I-TCX area colocalized with the core of ischemic parenchyma, whereas the I-PCX was close to the border between the ischemic and normal parietal cortex. The sampling region of I-ST was located slightly dorsolateral to the anterior commissure. The sampling area for the N-TCX, N-PCX, and N-ST were marked in the left hemisphere accordingly.

### 3.2. HILIC-MS/MS

Negative ion mode base peak chromatography of PLs and SLs in a normal TCX sample is exemplified in the Appendix A to illustrate the elution profile of lipid classes studied in this study. The Cer, PE, LPE, PC, SM, and LPC classes were sequentially eluted, with ample separation from the adjacent lipid classes.

### 3.3. Multivariate Analysis of PL and SL Information in the Brain Tissue

#### 3.3.1. Partial Least Square Discriminant Analysis (PLS-DA)

The tissue content of individual lipid species in N-TCX, N-PCX, N-ST, I-TCX, I-PCX, and I-ST were analyzed with LC-MS/MS, relative abundance calculated, then further analyzed with PLS-DA. The cumulative R2X (R2X(cum)) from PLS-DA was 0.813, and the overall cumulative cross-validated R2 (Q2(cum)) was 0.868. The scatter plot of this analysis is demonstrated in Figure 2, showing component 1 with explained variation (R2X[1]) of 0.375 and component 2 with explained variation (R2X[2]) of 0.206. The result indicated that the monitored lipids permitted clustering of individual tissue samples into groups according to their anatomical regions and their ischemic states. The N-ST group appeared well separated from the N-TCX and N-PCX groups, while the latter two did not separate well from each other. The I-TCX, I-PCX, and I-ST groups appeared well-separated from each other and located in the left half of the scatter plot, far away from the clusters of normal brain areas.

#### 3.3.2. Orthogonal Partial Least Square Discriminate Analyses (OPLS-DA), Volcano Plot Analyses, Modeling, and Validation of Lipid Markers

##### Normal versus Ischemic Brain Parenchyma

The scatter plot of PLS-DA on brain lipid samples indicated a clear separation between normal and ischemic brain tissue. Therefore, we performed the OPLS-DA on lipids of normal and ischemic brain parenchyma, using three samples each from TCX, PCX, and ST of normal and ischemic brain parenchyma to identify the ischemic brain markers. Figure 3A shows the scatter plot from this analysis, yielding a modeled cumulative variation R2 (i.e., R2(cum)) of 0.97, a cumulative X variation (R2X (cum)) of 0.647 that included a cumulative predictive X component (R2X[1] (cum)) of 0.269. The cumulative overall cross-validated variation of R2, i.e., the Q2(cum), was 0.88. The S-plot of this analysis identified the lipid species with a correlation coefficient score to the classification (p(corr)[1]) higher than 0.7 or lower than −0.7 as the potential marker species and revealed LPC 22:6, LPE 22:6, and LPE 20:4 together as the markers distinguishing the ischemic brain parenchyma from its normal counterpart (purple hexagons, Figure 3B). The volcano plot analysis of the same set of brain parenchyma lipids, using two folds of change (FC) (i.e., log2(FC) > 1 or <−1) and the *p* value of less than 0.05 (i.e., −log (p) > 1.301) as cutoffs, revealed the increase of LPC 22:6, LPE 22:6, LPE 20:4, PC 20:4/22:6, and LPE 18:0 and the decrease of PC 18:0/18:2, SM d18:1/16:0, SM d18:1/18:0, and SM d18:1/20:0 together as the lipid markers for the ischemic brain parenchyma (Figure 3C).

Table 1 listed all the lipid species showing statistical significance (*p* < 0.05) between ischemic and normal brain parenchyma with their respective t-statistic, *p* value, and the false discovery rate (FDR) (N = 9 each, unpaired *t*-test). The sum of FDR from all the lipid species listed in this table was 0.6766, whereas that from the lipid species identified by the S-plot of OPLS-DA was 0.0151, and that from the volcano-plot-identified lipid species was 0.2480.

To validate the S-plot- and volcano-plot-identified lipid markers, the multiple-sample receiver operating characteristic curve (MSROCC) models were established using the lipid information identified in each plot. Thereafter, the lipid information in the additional 11 TCX, 11 PCX, and 9 ST samples from the ischemic and normal brain parenchyma each were classified by these two MSROCC models. Confusion matrix calculation of the classification outcome yielded the accuracy, precision, sensitivity, and specificity of 1.0000 each, with the false positive rate and the false negative rate of 0.0000 each for S-plot-identified lipid markers. Similar calculation for the volcano-plot-identified lipid markers yielded an accuracy of 0.9839, precision of 1.0000, sensitivity of 0.9677, specificity of 1.0000, false positive rate of 0.0000, and false negative rate of 0.0323 (Table 2).

##### Temporal Cortex

The OPLS-DA of lipids in N-TCX and I-TCX was carried out to identify the differentiating lipid markers of normal temporal cortex from its ischemic counterpart. Figure 4A showed the scatter plot of this analysis, yielding a modeled R2(cum) of 0.998, an R2X (cum) of 0.758 that included an R2X[1] (cum) of 0.69, and a Q2(cum) of 0.993. The S-plot of this OPLS-DA analysis in Figure 4B revealed that the p(corr)[1] of LPE 18:0, LPC 16:0, LPE 16:0, LPC 20:4, LPE 20:4, LPC 22:6, PC 16:0/18:0, LPC 18:0, LPC 18:1, and LPE 22:6 were higher than 0.8 (upper-right purple hexagons and the upper-left inset, Figure 4B), whereas the p(corr)[1] of SM d18:1/18:0, PC 16:0/20:4, PC 18:0/20:4, PC 18:0/18:1, PE 16:0/22:6, Cer d18:1/22:0, PE 18:0/18:2, PC 16:0/18:1, PE 18:0/22:6, PC 18:0/18:2, PE 18:0/18:1, PC 16:0/22:6, PE 18:0/20:4, and PC 16:0/16:0 were lower than −0.8 (lower-left purple hexagons and lower-right inset, Figure 4B). The volcano plot identified the significant decrease of PC 16:0/22:6, PE 18:0/20:4, PC 18:0/18:2, PE 18:0/18:1, PE 18:0/18:2, Cer d18:1/22:0, PE 16:0/22:6, SM d18:1/18:0, and PE 16:0/18:1 together with the significant increase of LPE 18:0, LPC 16:0, LPE 16:0, LPE 20:4, LPC 20:4, LPC 22:6, LPC 18:0, and LPC 18:1, and Cer d18:1/18:1 served as the lipid markers differentiating the I-TCX from the N-TCX (Figure 4C).

Table 3 listed the t-statistic, *p* value, and FDR of lipids in the I-TCX that were statistically different from those in the N-TCX (unpaired *t*-test, N = 6 each). The sum of FDR from all lipids with *p* < 0.05 was 0.0819, whereas that from lipid markers identified by the S-plot in Figure 4B was 0.0074, and that from those identified by the volcano plot in Figure 4C was 0.0226.

The MSROCC models were again established based on the lipid marker identified by the S-plot and the volcano plot above, and the lipid data from the additional 8 N-TCX and 8 I-TCX tissue samples were classified by these two models. Confusion matrix calculation of the classification result yielded the accuracy, precision, sensitivity, and specificity of 1.0000 each, with the false positive rate and false negative rate of 0.000 each for S-plot- identified lipid markers. Similar calculation also revealed the accuracy of 0.9375, precision of 1.0000, sensitivity of 0.8750, specificity of 1.0000, false positive rate of 0.0000, and false negative rate of 0.1250 from volcano-plot-identified lipid markers (Table 2).

##### Parietal Cortex

Similar OPLS-DA was performed on lipids in N-PCX and I-PCX to identify the markers differentiating the normal parietal cortex from its ischemic counterpart. The scatter plot of this analysis in Figure 5A yielded a modeled R2(cum) of 0.994 and an R2X(cum) of 0.576, which included a predictive R2X[1] (cum) of 0.491. The cumulative Q2 (cum) of this analysis was 0.961. The S-plot of OPLS-DA on PCX lipids in Figure 5B identified PC 18:0/18:1, LPE 22:6, PC 20:4/22:6, and Cer d18:1/16:0, with their p(corr)[1] larger than 0.8 (upper-right hexagons and upper-left inset, Figure 5B), and SM d18:1/20:0, PC 16:0/18:1, PE 18:0/20:4, PC 16:0/22:6, PC 18:0/18:2, SM 18:0, PC 16:0/20:4, and PC 16:0/16:0 with their p(corr)[1] lower than −0.8 (lower-left hexagons and lower-right inset, Figure 5B). The volcano plot analysis of lipids in the same N-PCX and I-PCX samples identified the significant decrease of SM 18:0, PC 18:0/18:2, and SM d18:1/20:0, plus the significant increase of LPE 22:6, PC 20:4/22:6, Cer d18:1/16:0, Cer d18:1/24:0, LPC 20:4, LPC 22:6, LPC 18:1, and LPC 18:0 together as the differentiating markers of I-PCX from the N-PCX.

Table 4 listed the t-statistic, *p* value, and FDR of individual lipids in the I-TCX that were significantly different from their counterparts in the N-TCX (N = 6 each, unpaired *t*-test). The sum of FDR from all the lipid species on this table was 0.284, whereas that from S-plot-identified lipid markers was 0.0147, and that from volcano-plot-identified lipid markers was 0.0620.

Again, the MSROCC models based on the S-plot and the volcano-plot-identified lipid markers were established. Classification of an additional eight samples from N-PCX and I-PCX each was performed according to these two models. Confusion matrix calculation revealed the accuracy, precision, sensitivity, and specificity of 1.0000 each, with both false positive rate and false negative rate of 0.0000 for S-plot-identified lipid markers. Similar calculation on the classification result of volcano-plot-identified lipid markers yielded an accuracy of 0.9375, precision of 0.8889, sensitivity of 1.0000, specificity of 0.8750, false positive rate of 0.1250, and false negative rate of 0.0000 (Table 2).

##### Striatum

OPLS-DA of lipids in N-ST and I-ST resulted in the scatter plot shown in Figure 6A, yielding a modeled R2(cum) of 0.993, an R2X (cum) of 0.607 that included a predictive R2X[1] (cum) component of 0.512, and a cumulative Q2(cum) of 0.97. The S-plot of this analysis in Figure 6B identified LPE 22:6, PC 16:0/18:0, LPE 20:4, LPC 16:0, LPC 22:6, LPC 18:1, PC 20:4/22:6, LPE 18:1, PC 18:0/18:1, and PC 18:0/20:4 with their p(corr)[1] higher than 0.8 (upper-right purple hexagons and upper-left inset, Figure 6B), and SM d18:1/18:0, PE 16:0/18:1, PE 16:0/22:6, and SM d18:1/16:0 with their p(corr)[1] lower than −0.8 (lower-left purple hexagons and lower-right inset). The volcano plot analysis of ST lipids from the same sample set identified the significant decrease of SM d18:1/16:0, SM d18:1/18:0, and SM d18:1/24:1, and the significant increase of LPE 22:6, PC 16:0/18:0, LPC 16:0, LPC 22:6, LPE 20:4, LPC 18:1, LPE 18:1, PC 20:4/22:6, LPC 20:4, and LPE 18:0 together in I-ST could differentiate itself from the N-ST (Figure 6C).

Table 5 listed the t-statistic, *p* value, and the FDR of lipids in I-ST that were statistically different (*p* < 0.05) from their counterparts in N-ST (N = 6 each, unpaired *t*-test). The sum of FDR from all lipids on this table was 0.202, whereas that from S-plot-identified lipid markers was 0.0119, and that from volcano-plot-identified lipid markers was 0.0602.

The MSROCC models based on the S-plot and volcano-plot-identified lipid markers were established again. Classification of an additional six samples from N-ST and I-ST each were performed using these two models. Confusion matrix calculation yielded the accuracy, precision, sensitivity, and specificity of 1.0000 each, with both false positive rate and false negative rate of 0.0000 for S-plot-identified lipid markers. The same calculation on the classification result by volcano-plot-identified lipid markers reached the accuracy, precision, sensitivity, and specificity of 1.0000, and both false positive rate and false negative rate of 0.0000 (Table 2).

## 4. Discussion

In this study we examined the common PLs and SLs in the ischemic brain parenchyma and compared their tissue levels with those in the contralateral nonischemic brain tissue at TCX, PCX, ST, and the brain parenchyma as well. We identified the ischemic lipid markers using OPLS-DA and volcano plot analyses for each anatomical region. Classification of additional tissue cohorts based on the MSROCC models was performed, and the accuracy, precision, sensitivity, specificity, false positive rate, and false negative rate of each lipid marker panel was calculated. In general, the OPLS-DA-identified lipid marker panels contained fewer lipid species yet resulted in a lower sum of false discovery rates and yielded better accuracy, precision, sensitivity specificity, false positive rate, and false negative rate in the validating classification of additional tissue cohorts.

The OPLS-DA result revealed one polyunsaturated fatty acyl (PUFA) lyso-PC (LPC 22:6) and two PUFA lyso-PEs (LPE 20:4 and LPE 22:6; Table 1) as the lipid markers differentiating the ischemic brain parenchyma from the normal brain tissue, the combination of which yields a sum of false discovery rate of less than 2% (0.0151). The volcano plot analysis, on the other hand, included additional PL and SL species in addition to these three PUFA-PLs yet resulted in an unacceptable false discovery rate of almost 25% (0.2480). Due to the wide variety of anatomic natures of brain tissue included in this OPLS-DA analysis, the cutoff of the coefficient score to the classification (p(corr)[1]) of 0.7 or −0.7 was adapted instead of 0.8 or −0.8 used in the rest of this study to accommodate the differentiation between ischemic and normal brain tissue. Nevertheless, the result still yielded a relatively reliable classification outcome. The three lyso-PLs identified herein were also identified as the ischemic lipid markers in TCX and ST yet only partially in the PCX by OPLS-DA. Such a minor discrepancy was likely attributable to the overall data weighing in the sample of OPLS-DA. Alternatively, the I-PCX sampling region was close to the normal-ischemia border (Figure 1) that was generally regarded as the ischemic penumbra not as severely compromised by ischemia. Therefore, presenting a set of lipid markers of slight difference may merely reflect the pathophysiological state of the sampling area.

In our previous MALDI-MSI study of lipids in the ischemic rat brain parenchyma, we reported a small yet significant increase of PC 16:0/18:0 in addition to the significant decrease of several PCs, SM, and the significant increase of LPC 16:0 [15]. In this quantitative study, we confirmed such increase of PC 16:0/18:0 in all three ischemic brain regions (Table 3, Table 4 and Table 5). In addition, the increase of PC 20:4/22:6 was also observed in the PCX and ST but not in the TCX. Such aberrant increase of rare PCs in the ischemic brain tissue was likely the summarizing outcome of upregulation in Lands’ cycle [27] mediated through heightened activity of lysophosphatidylcholines acyl transferases (LPCATs) [28]. The absence of PC 20:4/22:6 in the TCX might be attributed to the severity of tissue and cellular damage sustained by this core ischemic brain region that compromised the reacylation of lyso-PCs. Nevertheless, detailed up- and downregulations of LPCATs involved in the increase of such rare rat brain lipid species in the ischemic brain await further investigation.

The identities of OPLS-DA-revealed lipid markers in the ischemic brain parenchyma were significantly different from those reported previously [14,16,29,30]. In those earlier studies, the MALDI matrix of selection was either 2,5-dihydroxy benzoic acid (DHB) or 9-aminoacridine (9-AA). Neither of these matrices offered advantages in MALDI ionization of PL classes such as PEs that were unambiguously detectable only by the use of an unconventional matrix such as graphene oxide [31] or by the combined oversampling and laser postionization approach [32]. Hence, the sampling bias in those studies inadvertently led to the conclusion that excluded the most common PL classes, such as PEs and LPEs. An obvious example was seen in the OPLS-DA-identified, upregulated lipids in the I-TCX. Other than the upregulated PC 16:0/18:0, 9 LPLs accounted for all of the rest of the 10 upregulated lipids, and LPEs accounted for 4 of these 9 LPLs. In brain regions less affected by ischemia, such as the ischemic penumbra of I-PCX, the number of upregulated lipids was reduced to four species, and LPE 22:6 was the only LPL identified. This strongly suggests that an increase of LPEs in brain parenchyma may serve as a more sensitive marker of suspected brain ischemia than LPCs could offer, and the number of significantly upregulated LPEs in tissue could reflect the severity of ischemia sustained by the local brain tissue. However, due to a limitation in detected lipid classes, lack of discussion on the possible functionality of LPEs in the ischemic brain tissue has strongly undermined its significance as a class of versatile markers. Nevertheless, in this study we adapted a different ionization approach suitable for ionizing an individual PL and SL class at a different elution segment so as to provide a more comprehensive lipid species information and to avoid sampling bias. As the pool of detected lipid species expands, reaching a conclusion of different differentiating lipid markers of brain ischemia from those by MALDI studies should be no surprise. However, as the sensitivity and mass resolution of the lipid-detecting mass spectrometer system improves beyond that used in this study, the number of detected lipid species and the quality of their ion signals will increase; further deviation of the ischemic lipid markers from the current conclusion may also be reached inadvertently.

In conclusion, in this study we identified and validated region-specific lipid markers in the ischemic rat brain parenchyma. The identities of these lipid markers appear largely different from those reported by MALDI methods, mostly due to the inclusion of additional candidates of marker lipid classes and a more reliable relative quantitative information. LPEs may serve more effectively as the qualitative and quantitative tissue lipid markers of ischemia to monitor the ischemic assault to the brain parenchyma and to gauge the extent of ischemia-mediated tissue damage. The information herein will bring forth additional insights to lipid marker studies and to tissue marker studies in general.

## Figures and Tables

**Figure 1 metabolites-12-01075-f001:**
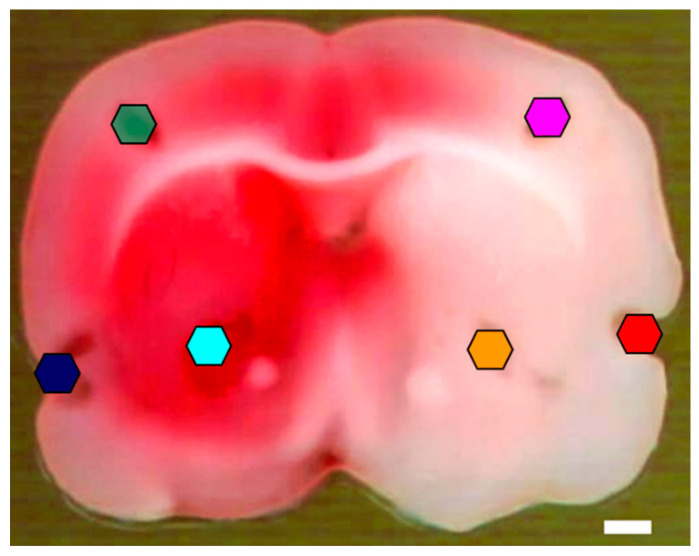
TTC stain of a coronal rat brain section after tissue puncture. The normal brain area at the left hemisphere was stained in red, whereas the ischemic brain area was stained in white. Punctured areas were color-marked in hexagons: blue: normal temporal cortex (N-TCX); green: normal parietal cortex (N-PCX); light blue: normal striatum (N-ST); red: ischemic temporal cortex (I-TCX); pink: ischemic parietal cortex (I-PCX); orange: ischemic striatum (I-ST). Color codings of the specific brain regions in this figure are used throughout. Bar = 1 mm.

**Figure 2 metabolites-12-01075-f002:**
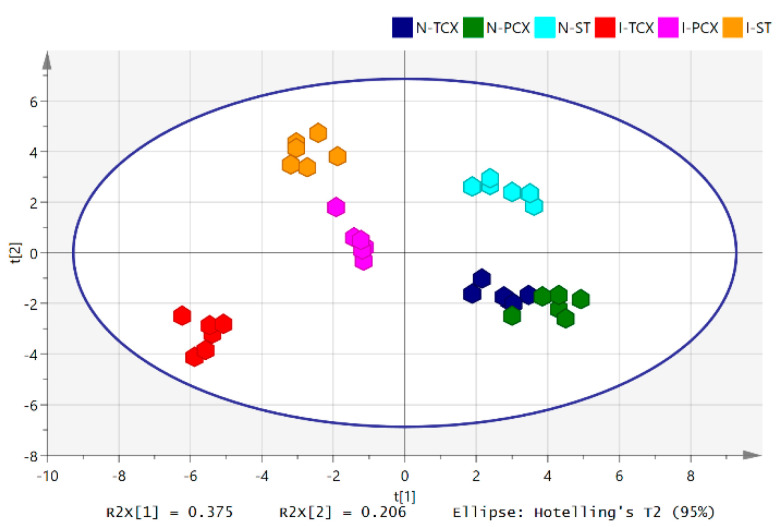
Scatter plot of partial least square discriminating analysis (PLS-DA) on phospholipids and sphingolipids in the ischemic and normal brain parenchyma. N = 6 at each sampling area. Color coding of the sampling regions follows that in Figure 1.

**Figure 3 metabolites-12-01075-f003:**
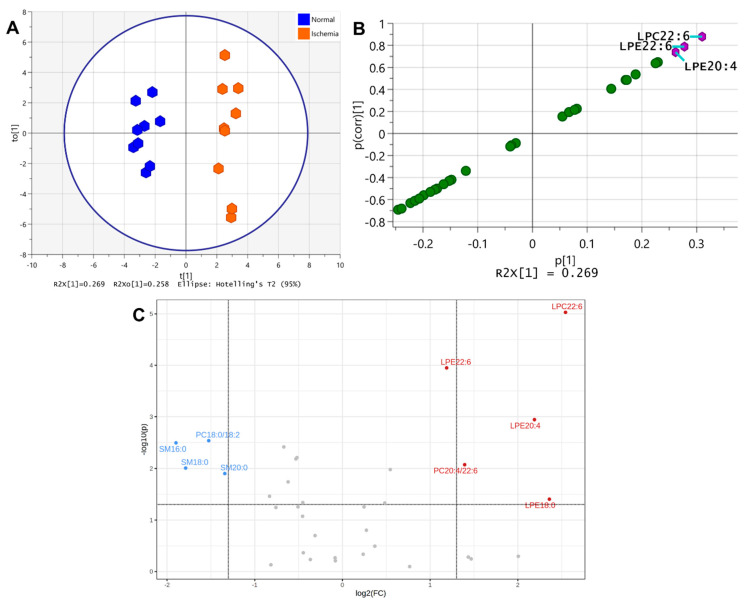
(**A**): The scatter plot of OPLS-DA on lipids in normal and ischemic rat brain parenchyma. (**B**): S-plot of OPLS-DA on lipids in normal and ischemic rat brain parenchyma. Lipid species with p(corr)[1] > 0.7 or <−0.7 are identified (purple hexagons). (**C**): Volcano plot analysis on lipids in normal and ischemic rat brain parenchyma. Horizontal dotted line denotes −log10(p) of 1.301. Left vertical dotted line denotes log2(FC) of −1.322 (i.e., FC = −2.5; FC: fold change), and the right vertical dotted line denotes log2(FC) of 1.322 (i.e., FC = 2.5) Lipids with −log10(p) > 1.301 and log2(FC) > 1 or <−1 are identified. See text for details.

**Figure 4 metabolites-12-01075-f004:**
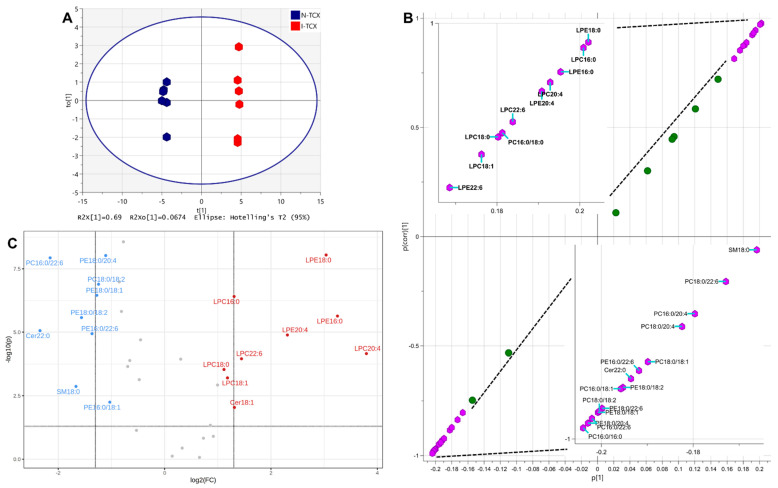
(**A**): The scatter plot of OPLS-DA on lipids in normal temporal cortex (N-TCX) and ischemic temporal cortex (I-TCX) of rats. (**B**): S-plot of OPLS-DA on lipids in the N-TCX and the I-TCX of rats. Lipid species with p(corr)[1] > 0.8 (upper-right purple hexagons and upper-left inset) or <−0.8 (lower-left purple hexagons and lower-right inset) are identified. (**C**): Volcano plot analysis of lipids in normal and ischemic rat brain parenchyma. Lipids with −log10(p) > 1.301 and log2(FC) > 1 or <−1 are identified. See Figure 3 legend and text for details.

**Figure 5 metabolites-12-01075-f005:**
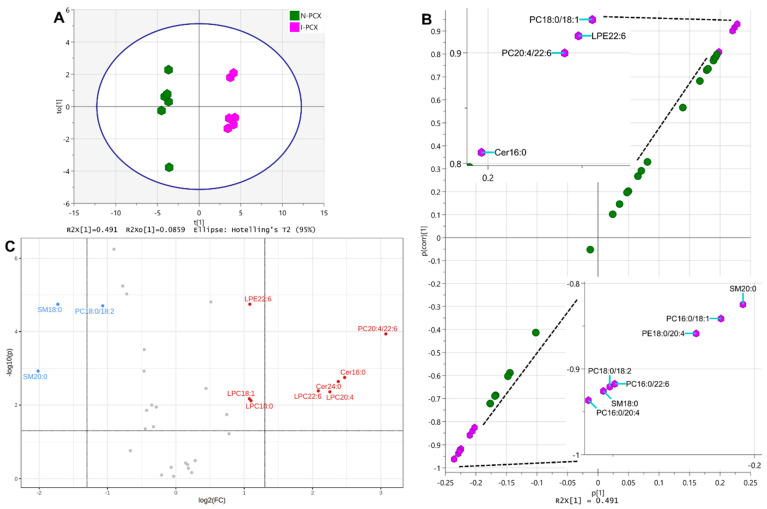
(**A**): The scatter plot of OPLS-DA on lipids in normal parietal cortex (N-PCX) and ischemic parietal cortex (I-PCX) of rat brains. (**B**): S-plot of OPLS-DA on lipids in N-PCX and I-PCX of rat brains. Lipid species with p(corr)[1] > 0.8 (upper-right purple hexagons and upper-left inset) or <−0.8 (lower-left purple hexagons and lower-right inset) are identified. (**C**): Volcano plot analysis on lipids in N-PCX and I-PCX of rat brains. Lipids with −log10(p) > 1.301 and log2(FC) > 1 or <−1 are identified. See Figure 3 legend and text for details.

**Figure 6 metabolites-12-01075-f006:**
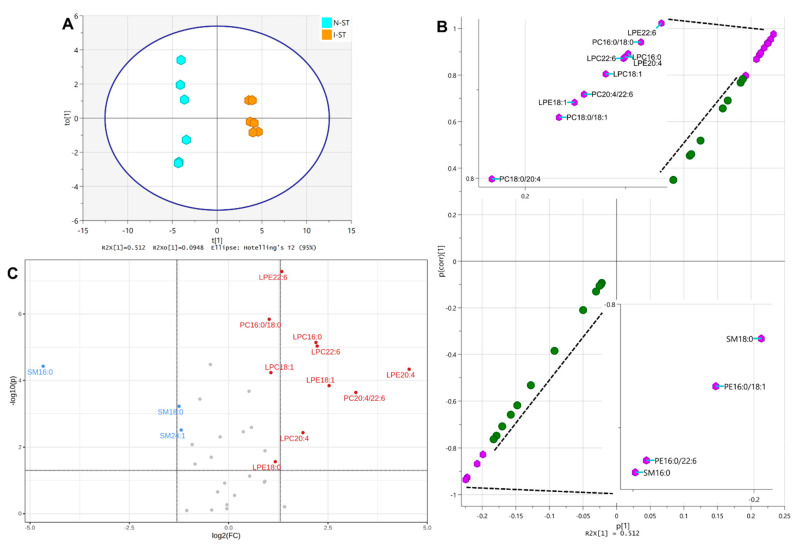
(**A**): The scatter plot of OPLS-DA on lipids in normal striatum (N-ST) and ischemic striatum (I-ST) of rat brains. (**B**): S-plot of OPLS-DA on lipids in N-ST and I-ST of rat brains. Lipid species with p(corr)[1] > 0.8 (upper-right purple hexagons and upper-left inset) or <−0.8 (lower-left purple hexagons and lower-right inset) are identified. (**C**): Volcano plot analysis on lipids in N-ST and I-ST of rat brains. Lipids with −log10(p) > 1.301 and log2(FC) > 1 or <−1 are identified. See Figure 3 and text for details.

**Table 1 metabolites-12-01075-t001:** List of lipid species showing statistical significance (*p* < 0.05) between ischemic and normal brain parenchyma (N = 9 each, unpaired *t*-test), with their t-statistic, *p* value, −log(p), false discovery rate (FDR) of each listed lipid species, the sum of FDR from all listed lipid species, (the FDR column), from s-plot of OPLS-DA-identified markers, and from volcano-plot-identified markers.

Normal vs. Ischemic Brain Parenchyma						
Lipid Species	t-Statistic	*p* Value	−Log (p)	FDR	FDR of Lipid Markers by S-Plot	FDR of Lipid Markers by Volcano Plot
LPC18:1	2.1542	0.0468	1.3296	0.093625		
LPC22:6	6.3667	9 × 10^−6^	5.0293	0.000318	0.000318	0.000318
LPE18:0	2.2424	0.0395	1.4039	0.089423		0.089423
LPE20:4	3.9538	0.0011	2.9441	0.012889	0.012889	0.012889
LPE22:6	5.0757	0.0001	3.9493	0.001910	0.001910	0.001910
PC16:0/16:0	−3.376	0.0039	2.4145	0.021820		
PC16:0/18:0	2.8979	0.0105	1.9793	0.032414		
PC16:0/18:1	−3.1538	0.0061	2.2114	0.027730		
PC16:0/22:6	−2.3093	0.0346	1.4608	0.084052		
PC18:0/18:2	−3.5085	0.0029	2.5359	0.021820		0.0218
PC20:4/22:6	3.0017	0.0085	2.0731	0.031922		0.0319
PE16:0/22:6	−2.6302	0.0182	1.7402	0.047575		
PE18:0/20:4	−3.1253	0.0065	2.1854	0.027730		
PE18:0/22:6	−2.1648	0.0459	1.3385	0.093625		
SM16:0	−3.4618	0.0032	2.4932	0.021820		0.021820
SM18:0	−2.9298	0.0098	2.0081	0.032414		0.032414
SM20:0	−2.8117	0.0125	1.9019	0.035516		0.035516
Sum of FDR				0.6766	0.0151	0.2480

**Table 2 metabolites-12-01075-t002:** Accuracy (Accu), precision (Prec), sensitivity (Sens), specificity (Spec), false positive rate (FPR), and false negative rate (FNR) of validation result using S-plot and volcano-plot-identified lipid markers from normal vs ischemic brain parenchyma (Normal vs. Ischemia), N-TCX vs. I-TCX, N-PCX vs. I-PCX, and N-ST vs. I-ST. See text for details.

	S-plot-identified Lipid Markers						Volcano-plot-identified Lipid Markers					
	Accu	Prec	Sens	Spec	FPR	FNR	Accu	Prec	Sens	Spec	FPR	FNR
Normal vs. Ischemia	1.0000	1.0000	1.0000	1.0000	0.0000	0.0000	0.9839	1.0000	0.9677	1.0000	0.0000	0.0323
N-TCX vs. I-TCX	1.0000	1.0000	1.0000	1.0000	0.0000	0.0000	0.9375	1.0000	0.8750	1.0000	0.0000	0.1250
N-PCX vs. I-PCX	1.0000	1.0000	1.0000	1.0000	0.0000	0.0000	0.9375	0.8889	1.0000	0.8750	0.1250	0.0000
N-ST vs. I-ST	1.0000	1.0000	1.0000	1.0000	0.0000	0.0000	1.0000	1.0000	1.0000	1.0000	0.0000	0.0000

**Table 3 metabolites-12-01075-t003:** List of lipid species showing statistical significance (*p* < 0.05) between I-TCX and N-TCX (N = 6 each, unpaired *t*-test), with their respective t-statistic, *p* value, −log(p), false discovery rate (FDR) of each listed lipid species, and the sum of FDR from all listed lipid species, (the FDR column), from the S-plot of OPLS-DA-identified markers, and from the volcano-plot-identified markers.

N-TCX vs. I-TCX						
Lipid Species	t-Statistic	*p* Value	−Log (p)	FDR	FDR of Lipid Markers by S-Plot	FDR of Lipid Markers by Volcano Plot
Cer18:1	3.2239	0.009115	2.0402	0.011478		0.011478
Cer22:0	−8.2726	8.77 × 10^−6^	5.057	2.48 × 10^−5^	2.48 × 10^−5^	2.48 × 10^−5^
Cer24:0	2.2796	0.045819	1.339	0.055638		
LPC16:0	11.6320	3.91 × 10^−7^	6.4073	1.66 × 10^−6^	1.66 × 10^−6^	1.66 × 10^−6^
LPC18:0	5.4214	0.000292	3.534	0.000473	0.000473	0.000473
LPC18:1	4.8972	0.000626	3.2036	0.000967	0.000967	0.000967
LPC20:4	6.4867	7.01 × 10^−5^	4.1541	0.000149	0.000149	0.000149
LPC22:6	6.1361	0.000110	3.9573	0.000216	0.000216	0.000216
LPE16:0	9.5923	2.32 × 10^−6^	5.6336	7.90 × 10^−6^	7.90 × 10^−6^	7.90 × 10^−6^
LPE18:0	17.2700	8.96 × 10^−9^	8.0476	9.88 × 10^−8^	9.88 × 10^−8^	9.88 × 10^−8^
LPE20:4	7.9117	1.30 × 10^−5^	4.8869	3.15 × 10^−5^	3.15 × 10^−5^	3.15 × 10^−5^
LPE22:6	4.4734	0.001191	2.924	0.001687	0.001687	
PC16:0/16:0	−19.5290	2.71 × 10^−9^	8.5672	9.21 × 10^−8^	9.21 × 10^−8^	
PC16:0/18:0	6.1086	0.000114	3.9416	0.000216	0.000216	
PC16:0/18:1	−10.0410	1.53 × 10^−6^	5.815	5.78 × 10^−6^	5.78 × 10^−6^	
PC16:0/20:4	−5.6134	0.000224	3.6507	0.00038	0.00038	
PC16:0/22:6	−16.8140	1.16 × 10^−8^	7.9349	9.88 × 10^−8^	9.88 × 10^−8^	9.88 × 10^−8^
PC18:0/18:1	−7.5211	2.01 × 10^−5^	4.6961	4.56 × 10^−5^	4.56 × 10^−5^	
PC18:0/18:2	−13.0870	1.29 × 10^−7^	6.8904	7.29 × 10^−7^	7.29 × 10^−7^	7.29 × 10^−7^
PC18:0/20:4	−6.0154	0.000129	3.888	0.000232	0.000232	
PC18:0/22:6	−4.7915	0.000733	3.1349	0.001084	0.001084	
PE16:0/18:1	−3.5068	0.005662	2.2471	0.007404		0.007404
PE16:0/22:6	−8.0183	1.15 × 10^−5^	4.9377	3.02 × 10^−5^	3.02 × 10^−5^	3.02 × 10^−5^
PE18:0/18:1	−11.7640	3.52 × 10^−7^	6.4533	1.66 × 10^−6^	1.66 × 10^−6^	1.66 × 10^−6^
PE18:0/18:2	−9.4471	2.67 × 10^−6^	5.5734	8.25 × 10^−6^	8.25 × 10^−6^	8.25 × 10^−6^
PE18:0/20:4	−17.1760	9.45 × 10^−9^	8.0245	9.88 × 10^−8^	9.88 × 10^−8^	9.88 × 10^−8^
PE18:0/22:6	−13.3840	1.04 × 10^−7^	6.983	7.07 × 10^−7^	7.07 × 10^−7^	
SM18:0	−4.3894	0.001357	2.8673	0.001846	0.001846	0.001846
Sum of FDR				0.0819	0.0074	0.0226

**Table 4 metabolites-12-01075-t004:** List of lipid species showing statistical significance (*p* < 0.05) between I-PCX and N-PCX (N = 6 each, unpaired *t*-test), with their t-statistic, *p* value, −log(p), false discovery rate (FDR) of each listed lipid species, the sum of FDR from all listed lipid species, (the FDR column), from S-plot of OPLS-DA-identified markers, and from volcano-plot-identified markers.

N-PCX vs. I-PCX						
Lipid Species	t-Statistic	*p* Value	−Log (p)	FDR	FDR of Lipid Marker by S-Plot	FDR of Lipid Marker by Volcano Plot
Cer16:0	4.2181	0.001777	2.7503	0.005035	0.005035	0.005035
Cer24:0	4.0590	0.002290	2.6401	0.005990		0.005990
LPC18:0	3.3297	0.007622	2.1179	0.014397		0.014397
LPC18:1	3.4011	0.006758	2.1702	0.013516		0.013516
LPC20:4	3.6626	0.004370	2.3595	0.009287		0.009287
LPC22:6	3.6989	0.004116	2.3856	0.009287		0.009287
LPE18:1	2.8200	0.018159	1.7409	0.028064		
LPE22:6	7.6249	1.79 × 10^−5^	4.7475	9.59 × 10^−5^	9.59 × 10^−5^	9.59 × 10^−5^
PC16:0/16:0	−11.191	5.62 × 10^−7^	6.2506	1.91 × 10^−5^	1.91 × 10^−5^	
PC16:0/18:0	3.7907	0.003539	2.4511	0.008596		
PC16:0/18:1	−4.4726	0.001193	2.9235	0.003686	0.003686	
PC16:0/20:4	−8.6856	5.69 × 10^−6^	5.2448	9.59 × 10^−5^	9.59 × 10^−5^	
PC16:0/22:6	−8.2117	9.36 × 10^−6^	5.0287	9.59 × 10^−5^	9.59 × 10^−5^	
PC18:0/18:1	7.7529	1.55 × 10^−5^	4.8101	9.59 × 10^−5^	9.59 × 10^−5^	
PC18:0/18:2	−7.5378	1.98 × 10^−5^	4.7044	9.59 × 10^−5^	9.59 × 10^−5^	9.59 × 10^−5^
PC18:0/20:4	−2.3739	0.039018	1.4087	0.057679		
PC20:4/22:6	6.0987	0.000116	3.936	0.000493	0.000493	0.000493
PE16:0/18:1	−2.9743	0.013943	1.8557	0.022574		
PE16:0/22:6	−3.0933	0.011379	1.9439	0.019345		
PE18:0/18:2	−2.2984	0.044375	1.3529	0.062864		
PE18:0/20:4	−5.3838	0.000308	3.5108	0.001165	0.001165	
PE18:0/22:6	−3.1678	0.010025	1.9989	0.017940		
SM18:0	−7.6222	1.79 × 10^−5^	4.7462	9.59 × 10^−5^	9.59 × 10^−5^	9.59 × 10^−5^
SM20:0	−4.4753	0.001188	2.9253	0.003686	0.003686	0.003686
Sum of FDR				0.2842	0.0147	0.0620

**Table 5 metabolites-12-01075-t005:** List of lipid species showing statistical significance (*p* < 0.05) between I-ST and N-ST (N = 6 each, unpaired *t*-test), with the t-statistic, *p* value, −log(p), false discovery rate (FDR) of each listed lipid species, and the sum of FDR from all listed lipid species, (the FDR column), from S-plot of OPLS-DA identified markers, and from volcano-plot-identified markers.

N-ST vs. I-ST						
Lipid Species	t-Statistic	*p* Value	−Log (p)	FDR	FDR of Lipid Marker by S-Plot	FDR of Lipid Markers by Volcano Plot
Cer22:0	−3.2757	0.008349	2.0784	0.014941		
LPC16:0	8.4571	7.21 × 10^−6^	5.1418	7.89 × 10^−5^	7.89 × 10^−5^	7.89 × 10^−5^
LPC18:0	3.0214	0.012864	1.8906	0.021869		
LPC18:1	6.6302	5.85 × 10^−5^	4.2326	0.000249	0.000249	0.000249
LPC20:4	3.7674	0.003677	2.4345	0.007354		0.007354
LPC22:6	8.2199	9.28 × 10^−6^	5.0326	7.89 × 10^−5^	7.89 × 10^−5^	7.89 × 10^−5^
LPE18:0	2.5818	0.027330	1.5634	0.042237		0.042237
LPE18:1	5.936	0.000144	3.8419	0.000544	0.000544	0.000544
LPE20:4	6.8193	4.63 × 10^−5^	4.3345	0.000225	0.000225	0.000225
LPE22:6	14.379	5.25 × 10^−8^	7.2802	1.78 × 10^−6^	1.78 × 10^−6^	1.78 × 10^−6^
PC16:0/18:0	10.106	1.44 × 10^−6^	5.8409	2.45 × 10^−5^	2.45 × 10^−5^	2.45 × 10^−5^
PC16:0/22:6	3.8079	0.003441	2.4633	0.007313		
PC18:0/18:1	5.6514	0.000212	3.6736	0.000712	0.000712	
PC18:0/18:2	−2.7602	0.020122	1.6963	0.032578		
PC18:0/20:4	3.9916	0.002553	2.593	0.006199	0.006199	
PC20:4/22:6	5.5914	0.000230	3.6374	0.000712	0.000712	0.000712
PE16:0/18:1	−5.268	0.000364	3.4391	0.001031	0.001031	
PE16:0/22:6	−7.0886	3.34 × 10^−5^	4.4762	0.000213	0.000213	
PE18:0/18:2	−2.4867	0.032172	1.4925	0.047558		
PE18:0/20:4	−3.5899	0.004930	2.3071	0.009313		
SM16:0	−6.991	3.76 × 10^−5^	4.4252	0.000213	0.000213	0.000213
SM18:0	−4.9257	0.000600	3.222	0.001569	0.001569	0.001569
SM24:1	−3.8796	0.003061	2.5142	0.006938		0.006938
Sum of FDR				0.2020	0.0119	0.0602

## Data Availability

The data presented in this study are available upon reasonable request from the corresponding author.

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
