# Peer review of "Quantitative Analyses and Validation of Phospholipids and Sphingolipids in Ischemic Rat Brains"

_metabolites, 2022, doi:10.3390/metabo12111075_

Round 1

Reviewer 1 Report

The authors, having experience in MALDI-MSI, apply their expertise to the quantitative analysis and validation of phospholipids and sphingolipids using ischemia for this purpose by HILIC-MS. During review, the following concerns arose:

1.    Would not it be more appropriate to use deuterium labelled lipids standards, which would be more similar to those found in the sample and therefore more accurate quantifications.

2.    The authors use brain tissue to analyse lipid composition. How do you normalise the analyses between the different samples? I have not seen any indication in the manuscript. In our case it is per μg of protein in each sample. But in other cases, I have seen it by weight of tissue extracted.

Author Response

We thank the reviewer's comments on our manuscript.  Here are our point-by-point response to reviewer's question/comment:

1. For LC-MS/MS based absolute quantitation of individual analyte, one should choose the internal standard having the same chemical characteristics to, yet distinguishable from the targeted analyte (see Ref. 22 of the manuscript).  Generally, the best option would be the use of a deuterium-labelled internal standard.  In addition, exogenous unlabeled analyte will also be needed for the generation of calibration curve for absolute quantitation. 

Theoretically it is possible to carry out absolute quantitation on mixture of lipids in tissue.  However, to acquire all 35 deuterium-labeled internal lipid standards and 35 corresponding unlabeled neat lipid standards would be practically impossible, since not many deuterium-labeled lipid internal standard species are commercially available.  Due to this practical limitation, we elected relative quantitation approach in which the internal standard for each lipid class has been confirmed to be absent in tissue, and the ratio of analyte to the internal standard of each lipid class was kept between 0.1 and 10 (See the second paragraph, section 2.4 of “Materials and Methods”[line 167-168], and Ref. 22). 

2. In section 2.3. “Tissue Lipid Extraction” of “Materials and Methods”, the following process was stipulated: “A 10µL aliquot of LIS containing 2.16 nmole of PE 14:0/14:0, 1.20 nmole of LPE 14:0, 2.16 nmole of PC 14:0/14:0, 1.08 nmole of SM d18:1/12:0, 1.73 nmole of LPC 14:0, and 1.81 nmole of ceramide d18:1/17:0 was added to the methanol/HCl mixture for every 3 mg of brain tissue.”  [line 125-128].  

Adding a fixed amount of internal standard for a fixed weight of tissue for homogenization, as the reviewer pointed out, was one way of normalization between samples.  We did describe such normalization approach in sample processing step. 

Reviewer 2 Report

This manuscript identified potential lipid markers that are region-specific in ischemic rat brain parenchyma.  The hypothesis is supported by a thorough analysis of molecular lipid species, especially PCs, LPCs, and SMs. The studies described by the authors in the manuscript contribute significantly to the existing knowledge of lipid markers in the brain.  

Author Response

We thank the reviewer’s comment and assurance on our study.  We have gone through Results of this manuscript and modified the description to improve its presentation.  In addition, we also found a few spelling error elsewhere and correct them as well.